# Parental competition for the regulators of chromatin dynamics in mouse zygotes

Masatoshi Ooga [1,2,3 ✉], Rei Inoue[2], Kousuke Kazama[2], Sayaka Wakayama [1], Satoshi Kamimura[1,4] & Teruhiko Wakayama [1]

The underlying mechanism for parental asymmetric chromatin dynamics is still unclear. To reveal this, we investigate chromatin dynamics in parthenogenetic, androgenic, and several types of male germ cells-fertilized zygotes. Here we illustrate that parental conflicting role mediates the regulation of chromatin dynamics. Sperm reduces chromatin dynamics in both parental pronuclei (PNs). During spermiogenesis, male germ cells acquire this reducing ability and its resistance. On the other hand, oocytes can increase chromatin dynamics. Notably, the oocytes-derived chromatin dynamics enhancing ability is dominant for the sperm-derived opposing one. This maternal enhancing ability is competed between parental pronuclei. Delayed fertilization timing is critical for this competition and compromises parental asymmetric chromatin dynamics and zygotic transcription. Together, parental competition for the maternal factor enhancing chromatin dynamics is a determinant to establish parental asymmetry, and paternal repressive effects have supporting roles to enhance asymmetry.

[1] Advanced Biotechnology Center, University of Yamanashi, Yamanashi 400-8510, Japan. [2] Faculty of Life and Environmental Sciences, University of Yamanashi, Yamanashi 400-8510, Japan. [3]Present address: Department of Animal Science and Biotechnology, School of Veterinary Medicine, Azabu University, Kanagawa 252-5201, Japan. [4]Present address: Stem Cell Biology Team, Institute for Quantum Life Science, National Institutes for Quantum Science and Technology, Chiba 263-8555, Japan. ✉email: ohga@azabu-u.ac.jp

During fertilization, parental pronuclei (PN) are formed from the genomes of the sperm and oocyte. Although co-existing in the cytoplasm of the zygote, the PNs separated before the first mitotic cell cycle, and then the two haplotypes fuse to form the new individual genome. There are many differential points in epigenetic factors (e.g., histone modifications, histone variants, and chromatin dynamics) between PNs before fusion. However, the mechanisms underlying parental asymmetry and its role remain unclear, and even less is known whether it is controlled by the interactions between the parental PNs.

The results of our previous study revealed that the chromatin of the male PN derived from the sperm (sp-mPN) was comparatively more dynamic among the preimplantation embryonic stages[1]. Interestingly, the chromatin of the female PN (fPN) is significantly less dynamic than that of the sp-mPN. Thus, the dynamics of the zygotic parental chromatin structure are asymmetrical. In addition to chromatin dynamics, the parental chromatin is asymmetric with size ($\male > \female$)[2], transcriptional regulation and activity ($\male > \female$)[3], epigenetic active and repressive of histone markers ($\male < \female$)[4] and the amounts of reprogramming factors ($\male > \female$)[5]. Round spermatids are haploid precursor cells present during the spermatogenetic stage soon after meiosis. Importantly, round spermatid injection (ROSI) and delay of intracytoplasmic sperm injection (ICSI) can result in improper PN formation and subsequent developmental failure during the preimplantation stage[6], suggesting the importance of parental asymmetry. However, the mechanism underlying parental asymmetry has not yet been elucidated. In addition, it is unknown whether the dynamics of the asymmetrical parental chromatin is due to an acquisition of a greater extent of histone mobility of the chromatin of the sp-mPN or decreased histone mobility in the fPN. Furthermore, the molecules involved in the regulation of the asymmetrical parental chromatin dynamics have not yet been identified.

It has recently become clear that the spermatozoon also plays a role in regulating the embryonic chromatin structure. For example, sperm carry epigenetic factors responsible for the highly complex organization of the genome[7–9] and DNA/histone modification and RNA in the zygote[10,11], which were thought to be involved in regulating the establishment of the zygotic chromatin structure and contribute to the control of embryonic development[7,11–14]. In addition to the factors carried by sperm, differences in the pronuclear formation process are critical for establishing chromatin structures in the zygotes. Soon after fertilization, only the sperm genome undergoes protamine-histone replacement. This male genome-specific phenomenon possibly confers the chance to male genomes for contact by the maternally supplied factors[15]. Although the contribution of spermatozoa to the establishment of the zygotic chromatin structure has been widely investigated, it remains unknown not only whether the molecular properties of sperm are involved in establishing the extreme dynamics of the sp-mPN chromatin but also whether these factors are actively involved in establishing the asymmetric dynamics of the parental chromatin after fertilization.

Therefore, the present study aimed to reveal the mechanisms underlying the asymmetric dynamics of the parental chromatin. The results of this study revealed that sp-mPN harbored the ability to further decrease the chromatin dynamics of the fPN, resulting in parental asymmetric chromatin dynamics. In addition to the ability of the sperm to further decrease the chromatin dynamics, our results indicated that the parental PNs compete for maternal ability to enhance the chromatin dynamics. Thus, the chromatin dynamics of the zygote are regulated by the repressive effect derived from the sperm and the enhancing effect derived from oocytes. Finally, the resultant asymmetric chromatin dynamics were involved in the parental asymmetric transcriptional regulation. Hence, the asymmetrical chromatin dynamics and gene expression in the zygotes are established by the competition between the parental PNs.

## Results

**Sperm represses chromatin dynamics in both parental PNs.** We previously reported that asymmetric dynamics of the parental chromatin were established in the late zygotic stage of the embryo at 10–12 h postinsemination (hpi)[1]. Firstly, we confirmed the reproducibility of asymmetric chromatin dynamics in the parental chromatin ($\male > \female$) in zygotes obtained by in vitro fertilization (IVF) (Supplementary Fig. 1) and ICSI by zygotic fluorescence recovery after photobleaching (zFRAP), which revealed histone mobility (an indicator of chromatin dynamics)[1,16,17]. In addition, the mechanisms causing the asymmetric parental pattern were investigated by determining whether sp-mPN acquired the higher histone mobility or fPN obtained the lower histone mobility. To this end, the changes in histone mobility during the early to mid-zygotic stages were examined using parthenogenetically activated- and ICSI-derived zygotes. Although the histone mobility of the fPN was gradually decreased along with the development of the zygote, there was no significant change in histone mobility in the sp-mPN (Fig. 1a, b). Importantly, in the presence of sperm/sp-mPN, the histone mobility of the fPN was further decreased. As a result, the parental asymmetric pattern ($\male > \female$) was established by 8 hpi. The dependency of the decreased histone mobility in fPN on sp-mPN was confirmed by enucleation of the sp-mPN followed by immunostaining of histone 3 lysine 9 trimethylation (H3K9me3) as a marker of the fPN (Fig. 1c and Supplementary Fig. 2). Collectively, these findings suggest that the asymmetrical histone mobility in the parental PNs was established via the acquisition of the decreased histone mobility of the fPN by a mechanism dependent on the sperm/sp-mPN.

Next, in order to determine whether the mechanisms underlying sp-mPN-dependent decreasing histone mobility were also activated in the sp-mPN itself, 1PN-ICSI were constructed by ICSI with enucleated metaphase II (MII) oocytes (Fig. 1d). The histone mobility of a single sp-mPN was lower than a single fPN of parthenogenetic zygotes (1PN-spt-partheno), although both appeared similar, suggesting that the decreasing histone mobility mechanism of sp-mPN worked on its own (Fig. 1e, f and Supplementary Fig. 3a).

Higher eGFP-H2B expression levels may result in higher histone mobility. We compared eGFP-H2B expression levels between 1PN-ICSI and 1PN-spt-partheno and confirmed that the histone mobility was determined regardless of the expression level of eGFP-H2B (Supplementary Fig. 3b, c), suggesting that histone mobility reflects a chromatin state. Further confirmation by ICSI was conducted with the use of enucleated MII oocytes fertilized with two sperm (2sp) (i.e., 1$\male$ (2sp) and 2$\male$ (2sp) in Fig. 1g, h and Supplementary Fig. 4a), which showed that the additional sperm resulted in further decreasing of histone mobility in each sp-mPN (Fig. 1h and Supplementary Fig. 4b). In zygotes formed by ICSI with two sp-mPNs and an un-enucleated oocyte (2$\male$ + 1$\female$), two sp-mPNs were comparable to one fPN and disruption of the parental asymmetric pattern ($\male \fallingdotseq \female$). Importantly, there was no significant difference between the fPNs of 2$\male$ + 1$\female$ and that of 1$\male$ + 1$\female$ (ctrl), indicating that the histone mobility of the fPNs was already decreased to almost the possible limit even in the presence of only one sperm. On the other hand, to decrease the histone mobility of sp-mPN to this level, at least two sperm were needed. Thus, the sp-mPN exhibited innate resistance to the ability for decreasing histone mobility. The zygotes harboring two fPNs with a single sp-mPN (1$\male$ + 2$\female$) still exhibited the parental asymmetric pattern (Supplementary Fig. 5). Thus, the additional chromatin from the fPN failed to disrupt the parental asymmetric

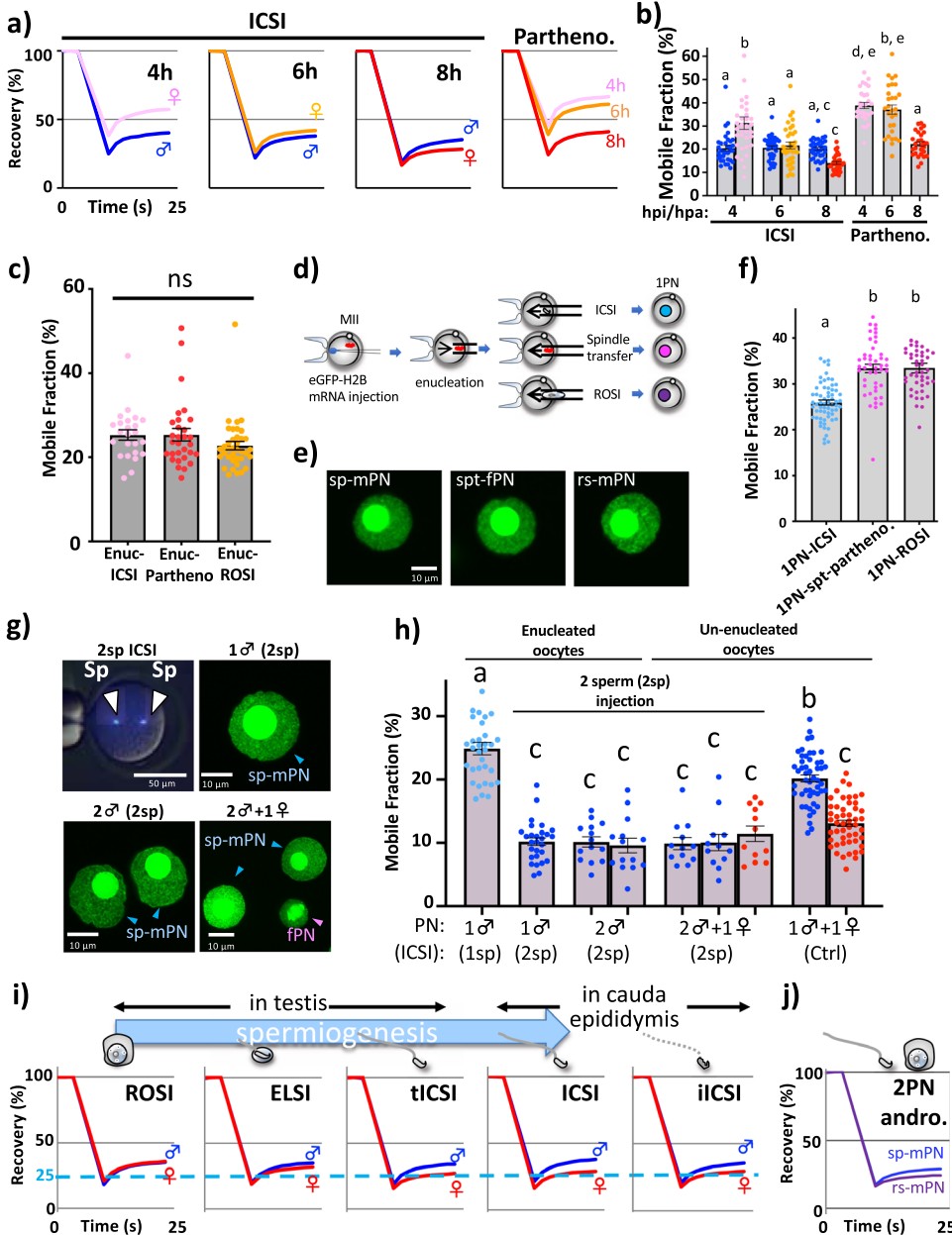

**Fig. 1 Sperm represses histone mobility in both parental PNs. a**, **b** Changes in histone mobility during the early and mid-stages of ICSI and parthenogenetically activated zygotes. zFRAP analysis was performed at 4, 6, and 8 hpi or hpa. A recovery curve indicating the average fluorescence recovery rate is shown (**a**). The average mobile fraction (MF) is shown as a gray bar (**b**). Single dots indicate the MF score of each male and female pronuclei (mPN, fPN), respectively. Blue, mPN; pink, fPN-4h; orange, fPN-6h; and red, fPN-8h. For parthenogenetic (partheno)-zygotes with 2PN, the average MF score is shown. Error bar indicates the standard error (SE). The examined number of embryos (n) = 34, 36, 35, 28, 29, 28, respectively. **c** mPN was enucleated at 4 hpi/hpa from ICSI- and ROSI-zygotes. Two PN partheno-zygotes were prepared as controls. The remaining fPN was subjected to zFRAP analysis at 8 hpi/hpa. n = 24, 37, 30, respectively. **d** Illustration of the preparation of 1PN-zygotes. **e** Fluorescence images of each single PN: sperm-derived mPN (sp-mPN), spindle transfer-derived fPN (spt-fPN), and round spermatid-derived mPN (rs-mPN). **f** Average MF scores of 1PN-ICSI, -ROSI, and -spt-partheno. n = 64, 45, 45, respectively. **g** Two sperm were injected into enucleated MII oocytes (upper-left). Sperm (sp) were stained with Hoechst 33342. Fluorescence images of PNs: 1♂ (2sp) and 2♂ (2sp) indicate the one and two male PN-zygotes injected with two sperm (2sp), respectively (upper right and lower left). Two sperm injected into un-enucleated MII oocytes (2♂ + 1♀; lower right). **h** Average MF scores of the zygotes shown in **g**. As a control, one sperm was injected into enucleated MII- (same as 1PN-ICSI in **f**; 1♂(1sp)) and normal ICSI-zygotes (1♂ + 1♀). Blue and red dots indicated mPN and fPN, respectively. n = 33, 28, 14, 12, 49, respectively. **i** Recovery curve of ROSI-, ELSI-, tICSI-, ICSI-, and iICSI-zygotes. n = 42,39, 34, 97, 47, respectively. **j** Recovery curve of androgenic zygotes prepared by co-injection of sperm and round spermatid. n = 17, 17, respectively. The error bar indicates SE.

pattern, indicating scarce or little repressive effect on histone mobility by fPN. Altogether, these results suggest that although the paternally repressive effect on histone mobility works on both parental PNs, parental asymmetric chromatin dynamics were established due to differences in sensitivity to this effect. Notably,

1PN ICSI and 1PN-spt-partheno have similar-sized PN, but they showed distinct histone mobilities (Fig. 1e, f). Furthermore, 1♂ (2sp) and 2♂ (2sp) zygotes whose PN sizes were different, showed similar histone mobility levels (Fig. 1g, h), indicating that histone mobility was determined regardless of PN size.

To examine where the paternal ability for the decreasing histone mobility in both parental PNs was acquired among the stages during spermiogenesis, zFRAP analysis of the zygotes fertilized by various micro-insemination methods (ROSI, ELSI, tICSI, and ICSI, see legend of Supplementary Fig. 6a) was performed at 8 hpi or hours post-activation (hpa). First, due to histone–protamine replacement during spermiogenesis, histone protein is drastically reduced in spermatozoa[18,19]. Therefore, it was possible that sp-mPN in ICSI-zygotes acquired the extremely higher amounts of eGFP-H2B and affected histone mobility. However, eGFP-H2B expression levels were not different between sp-mPN and rs-mPN (Supplementary Fig. 6b, c). Male PNs of zygotes obtained by ROSI and ICSI exhibited similar levels of histone mobility (Fig. 1i and Supplementary Fig. 6d). In contrast, the extent of histone mobility of the fPNs of these zygotes was decreased along with the maturity of the male germ cells (Fig. 1i and Supplementary Fig. 6d, e). Thus, these results suggested that paternal ability to repress histone mobility and its resistance acquired during spermiogenesis. Furthermore, fPN in ROSI-zygotes showed a higher histone mobility than ICSI-zygotes, indicating the scarce or little repressive effect on histone mobility by fPN.

During spermiogenesis, sperm acquires the ability to induce oocyte activation, which leads to the initiation of the zygotic cell cycle from meiotically arrested MII oocytes after fertilization. This ability can be easily inactivated by alkali treatment (Supplementary Fig. 6f)[20]. To examine whether this ability was involved in the parental asymmetric chromatin dynamics, histone mobility in the parental chromatin of the zygotes obtained by ICSI with the use of sperm that had been inactivated by alkali treatment (inactivated ICSI: iICSI) was only slightly decreased as compared to that of the ICSI-zygotes (Fig. 1i and Supplementary Fig. 6d, e), indicating that parental asymmetric chromatin dynamics acquired during spermiogenesis could not be explained by the activation capacity of the oocyte.

A variant of histone H3, H3.3, is considered important to establish open chromatin structure in parental PN after fertilization[21–23]. Therefore, we examined whether the differences of titration capacity of H3.3/Hira (H3.3 specific chaperon) were correlated with the chromatin dynamics in parental PNs of ROSI-, and ICSI-zygotes. However, there was no correlation between histone mobility and each level of endogenous or exogenous H3.3 and Hira, thereby indicating that it could not explain the levels of histone mobility in parental PNs in ROSI- and ICSI-zygotes by H3.3 level (Supplementary Fig. 7).

Correct discrimination of parental chromatin in ROSI-zygotes was confirmed by zFRAP analysis with maternal PNs from enucleated zygotes followed by immunocytochemical analysis of H3K9me3 as a marker of the fPNs (Supplementary Fig. 8). The inability of round spermatid to decrease histone mobility was confirmed by 1PN-ROSI, which showed a comparable level of histone mobility of 1PN-spt-partheno (Fig. 1f, purple). In addition, 2PN androgenic zygotes formed by sp-mPN, with comparatively greater histone mobility, and round spermatid-derived from the male PN (rs-mPN), with relatively less histone mobility (Fig. 1j and Supplementary Fig. 9), indicated that the round spermatid had not yet acquired resistance to the ability for decreasing histone mobility. Furthermore, in the presence of sp-mPN, the histone mobility of the rs-mPN was decreased to the same level as that of the fPN (Supplementary Fig. 10). These findings were consistent with the disruption of the asymmetric histone mobility of the parental PN in zygotes obtained by ROSI.

**Parental PNs compete for histone mobility-promoting factors.** As shown in Fig. 1f, h, the chromatin of 1PN-zygotes (namely, 1PN-spt-partheno and 1PN-ROSI) and even 1PN-ICSI-zygotes, was extremely dynamic as compared to that of ICSI-zygotes with two parental PNs (1♂ (1sp) vs. 1♂ + 1♀ (1sp)). The results of our previous study revealed that oocytes harbored high histone mobility and chromodomain helicase DNA binding protein 9 (CHD9) participated in the regulation of the histone mobility[24]. Furthermore, in another previous study, the transferred somatic cell nuclei into enucleated oocytes acquired a higher histone mobility, indicating the presence of factors promoting chromatin dynamics in oocytes and zygotes[1]. These findings prompted the hypothesis that the concentration of factors promoting histone mobility into a single PN led to the extremely dynamic chromatin structure. At the same time, such factors were distributed to the parental PNs. To examine this possibility, parthenogenetically activated oocytes were constructed with various numbers of fPNs (1, 2, and 4 fPNs; Fig. 2a), which enabled exclusion of the sp-mPN-derived the repressive effect on histone mobility. As expected, the extent of histone mobility decreased along with the increase in the number of PNs and the fPNs in the same zygotes showed a similar level of histone mobility (Fig. 2b and Supplementary Fig. 11). These results suggest that the chromatin dynamics promoting factors are present in the zygotes and at least, in parthenogenetic zygotes, the fPNs competed for these factors.

If the parental PNs compete for factors that promote histone mobility, the lack of one parental PN should cause excess histone mobility in another. Therefore, the potential of excess histone mobility in fPN was investigated with the use of enucleation of sp-mPNs (Fig. 2c). In sp-mPN-enucleated-zygotes, histone mobility of the fPN gradually increased along with the progression of zygotic development (Fig. 2d). In contrast, zFRAP analysis of ICSI-zygotes at 8, 10, and 11 hpi showed that the extent of histone mobility was maintained in both parental PNs. Thus, the parental PNs competed for the chromatin dynamics promoting factors. Next, the effect of delayed PN formation[6] on parental asymmetry was investigated. To this end, delayed ICSI-zygotes were constructed and then analyzed by zFRAP (Fig. 2e). Observation of PN formation of delay ICSI-zygotes revealed the reversal of PN size, larger or smaller, between parental PNs (Fig. 2f). zFRAP analysis revealed that asymmetric histone mobility of the parental chromatin was compromised along with an increased delay time of ICSI (Fig. 2g and Supplementary Fig. 12a). Particularly, almost all the 2 h-delayed ICSI-zygotes showed a reversed parental asymmetric pattern (♂ < ♀) (Supplementary Fig. 12b). A delay of only 1 h resulted in a considerable change in PN size and the fPN showed higher histone mobility than that of the sp-mPN. Collectively, these results indicate that parental PNs compete for histone mobility-promoting factors and the state of the zygotic chromatin is regulated by an antagonistic balance between the repressive effects on chromatin dynamics derived from the sperm and the promoting effect from the oocyte. Furthermore, it is possible that the sp-mPN might have obtained more such promoting factors than the fPN, resulting in self-resistance to the repressive effect. Notably, delaying fertilization timing was sufficient to reverse the parental asymmetry regardless of sperm deriving ability to decrease histone mobility, suggesting that the histone mobility-promoting factors were predominant in the regulation of parental asymmetric chromatin dynamics.

**Chromatin dynamics play a regulatory role for the establishment of parental asymmetric transcriptional activity.** High histone mobility is anticipated to be important in the zygotic transcriptional permissive state[25,26]. sp-mPN has higher transcriptional ability possibly due to a more transcriptional permissive chromatin state than fPN[15]. Thus, we hypothesized that

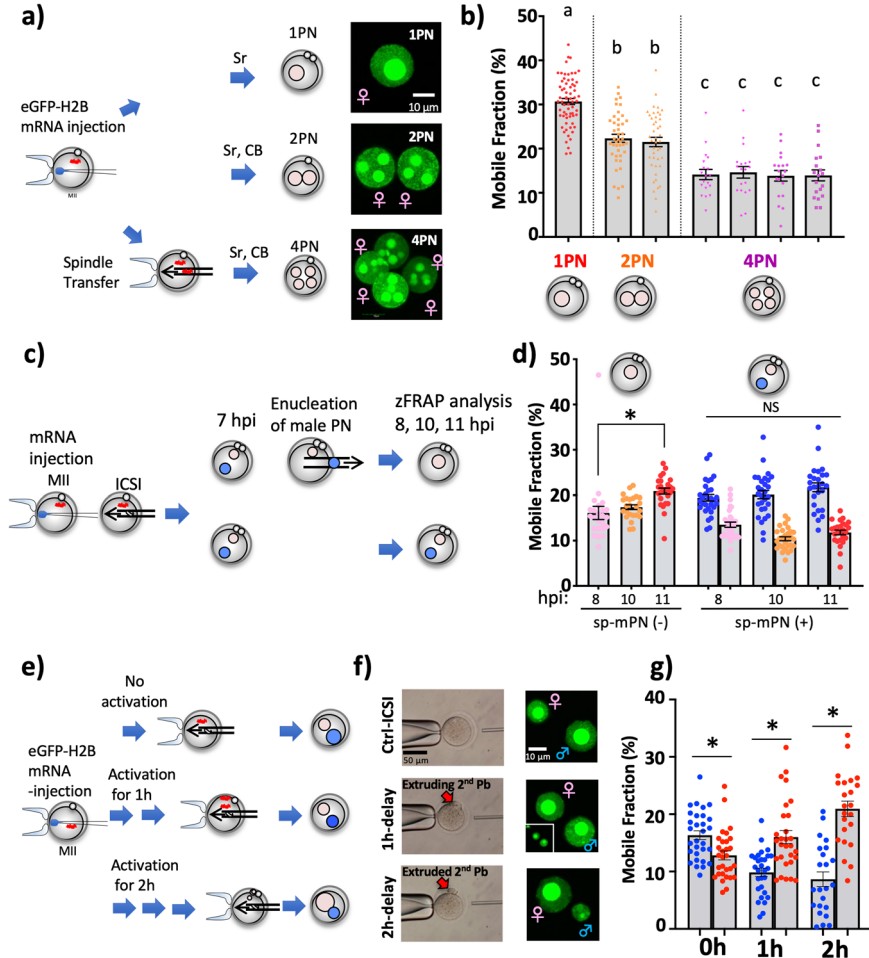

**Fig. 2 Parental PNs compete for chromatin dynamics promoting factors. a** Illustration of the preparation of 1, 2, and 4PN partheno-zygotes and fluorescence images of the fPNs. Sr indicates strontium, which induces oocyte activation. **b** Average MF scores of the partheno-zygotes prepared as shown in **a**. $n = 66, 43, 19$, respectively. **c** Illustration of the preparation of sp-mPN-enucleated zygotes. **d** Average MF scores of the zygotes prepared as shown in **c**. Blue, mPN; pink, fPN-8h; orange, fPN-10h; and red, fPN-11h). Asterisks indicate significant differences. Illustration of the preparation of delay ICSI-zygotes. $n = 24, 24, 26, 30, 29, 26$, respectively. **e** Illustration of the preparation of delay ICSI-zygotes. **f** Control and delay ICSI are shown. The second polar body is indicated with a red arrow. Fluorescence images of the mPN and fPN. The inset shows lower magnification images of the second polar body near the fPN. **g** Average MF scores of the delay ICSI-zygotes prepared as shown in **e**, **f**. $n = 30, 29, 23$, respectively. The error bar indicates SE.

higher histone mobility in sp-mPN is important for regulating parental asymmetric transcriptional activities. To examine this, we took advantage of our discovery that delayed ICSI induced the reversed parental asymmetric chromatin dynamics (Fig. 2e–g). As expected, the longer delay reduced transcriptional activity in sp-mPN, and eventually reversed parental asymmetric transcription was observed (Fig. 3a, b). Thus, reversed parental asymmetric chromatin dynamics reversed the parental asymmetric transcriptional activity, demonstrating that chromatin dynamics is positively correlated with the zygotic transcriptional activity. Considering that eviction and deposition of histone H2B could be coupled with transcription, it is possible that lowered transcriptional activity caused slower histone mobility. Nevertheless, the treatment with alpha-amanitin (ama), which is a RNA-polymerase II (Pol II) inhibitor, led to no reduction of histone mobility, indicating that histone mobility was not governed by transcriptional activity, but rather that chromatin dynamics regulate transcriptional activity (Fig. 3c). Importantly, in Fig. 2g, the chromatin dynamics were reversed only by a 1 h delay but in Fig. 3b it took as long as 2 h to completely reverse the parental asymmetry. These results suggest that chromatin dynamics positively regulates zygotic transcriptional activity, though it is not the sole factor. We further examined whether other RNA-

polymerase, RNA-polymerase I, and III (Pol I and Pol III), dependent transcriptions also did not regulate the establishment of parental asymmetric chromatin dynamics by using actinomycin D (Act D) and Pol III inhibitor (Pol IIIi). Act D could block not only Pol I but also DNA polymerase; however, in our experiment, DNA replication was not affected (Supplementary Fig. 13). Surprisingly, both inhibitors induced the reduction of histone mobility particularly in sp-mPN, suggesting that Pol I and Pol III mediated transcription caused in the higher histone mobility observed in sp-mPN (Fig. 3c).

## Discussion

In this study, the mechanisms underlying parental asymmetric chromatin dynamics were investigated, which revealed that it is determined by parental pronuclear competitive mechanisms. Sperm can reduce histone mobility of both parental PNs (Fig. 1a–h). The abilities to promote and resist the reduction of histone mobility are acquired during spermiogenesis (Figs. 1i, j and 3d). In addition to the sperm-derived ability to reduce histone mobility, oocytes also harbor factors that promote histone mobility, which the parental PNs compete for in zygotes (Fig. 2a–d). Notably, this competition is critical to the

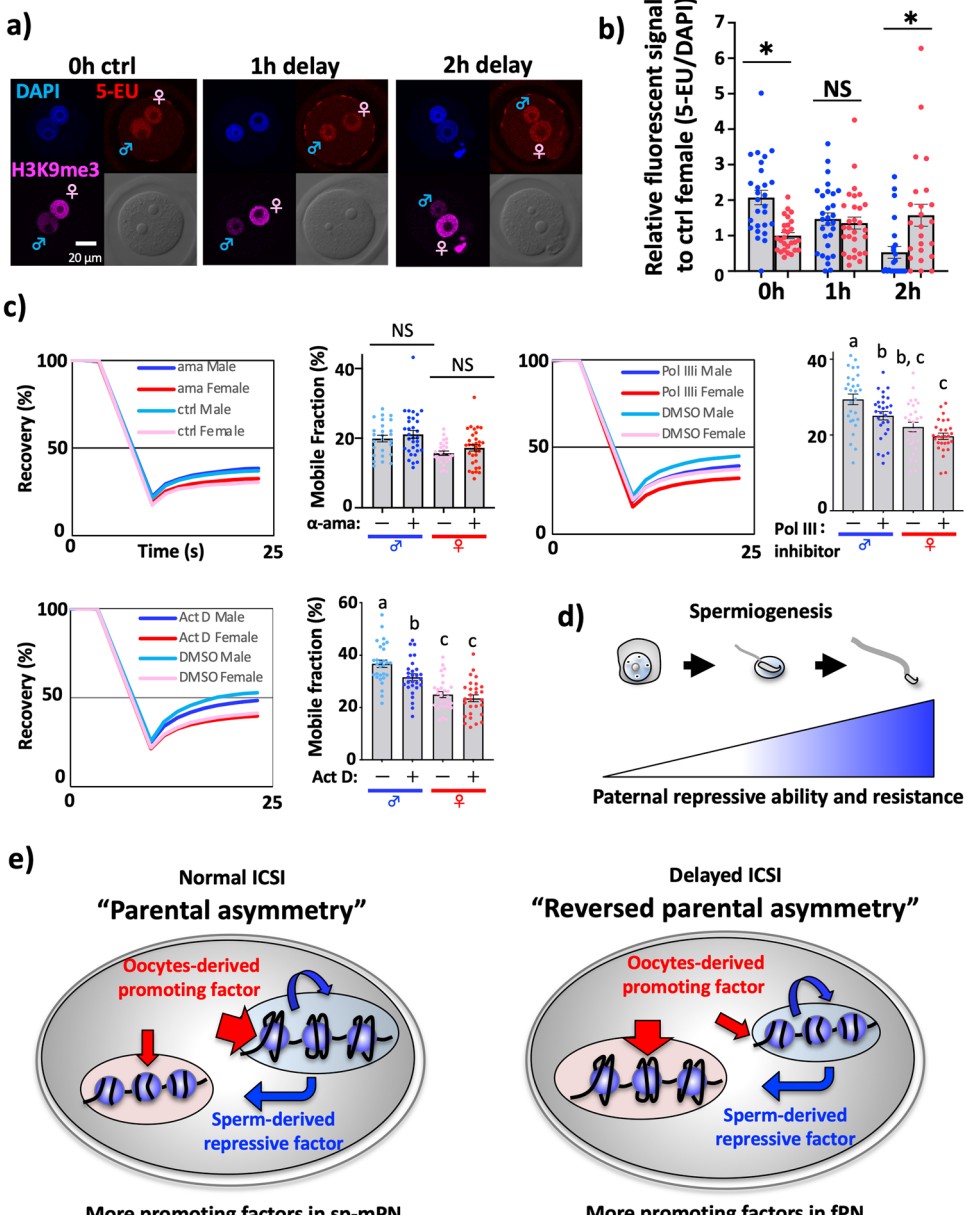

**Fig. 3 Chromatin dynamics play a regulatory role for the establishment of parental asymmetric transcriptional activity. a** Fluorescence images of 5-EU incorporation in delay ICSI-zygotes were shown. **b** Relative fluorescent signals were shown in the bar graph. Asterisks show significant differences by paired *t*-test. *n* = 27, 29, 24, respectively. **c** The average Recovery curve and MF scores of IVF- zygotes treated with 100-μg/ml α-amanitin, 0.1-μg/ml Act. D, 20-μM Pol IIIi and control zygotes were treated with 1 (Act. D) or 0.1% (Pol IIIi) dimethyl sulfoxide (DMSO) were shown. Treatment with these inhibitors started soon after mRNA injection (around 2hpi). zFRAP analysis was started at 8 hpi. *n* = 32, 26, 28, 29, 28, 28, respectively. **d** Schematic illustration, indicating male germ cells acquire the ability to repress chromatin dynamics and resistance during spermiogenesis. **e** In the zygotes, parental PNs compete for oocyte-derived histone mobility-promoting factors (red arrows) and the sperm-derived repressive factors (blue arrows), which are antagonistic. Probably, more histone mobility-promoting factors caused a more dynamic state and conferred resistance to reduce histone mobility in the sp-mPN. In the delayed ICSI-zygotes, more promoting factors in fPN caused reversed parental asymmetry. The error bar indicates SE.

establishment of parental asymmetric chromatin dynamics (Fig. 2e–g) and parental asymmetric transcriptional activity (Fig. 3a, b). Hereafter, this power balance is referred to as parental epigenetic competition.

It is probably that the differences in the dynamics of the parental PN formation process are involved in the regulation of parental epigenetic competition as a main underlying mechanism. The establishment of the chromatin structure of the sp-mPN is very distinct from that of the fPN. Within 1 h of fertilization, maternally pooled histone proteins are rapidly incorporated into the sp-mPN, resulting in the sperm head becoming decondensed

and expanded[27,28]. At this phase, maternal genetic materials still form completely condensed meiotic chromosomes, which are located in the cytoplasm or the going to be extruded second polar body (Fig. 2f)[29]. Also, it is widely thought that the transcription factors and chromatin remodeling factors dissociate from the condensed chromosomes and are re-recruited to the re-organized chromatin structure after chromosome segregation[30]. Therefore, it is likely that maternal factors were first taken up by the sp-mPN and then later by the fPN. In 1 h-delayed ICSI-zygotes, the asymmetric histone mobility of the parental chromatin structure was reversed (Fig. 2g and Supplementary Fig. 12). This

experiment was designed to collapse the competition for the maternally supplied factors from the ooplasm and resulted in the reversal of the parental asymmetry. Thus, it is possible that more maternally pooled or newly produced zygotic factors were incorporated into the sp-mPN than the fPN in normally fertilized zygotes. Indeed, more reprogramming factors that confer totipotency to the somatic cell nuclei and key transcriptional factors at this stage (e.g., SP1) are reportedly incorporated into the sp-mPN than the fPN[5,31]. Accordingly, it is plausible that more maternally supplied histone mobility-promoting factors could be incorporated into the sp-mPN than the fPN, resulting in asymmetric dynamics of the parental chromatin. In addition, changing the fertilization timing was sufficient to reverse the parental asymmetry regardless of the paternal repressive effects on chromatin dynamics suggests that the primary mechanism underlying parental asymmetry is the paternal PN having the advantage to start incorporating the chromatin dynamics promoting factors first. And paternal repressive effects have supporting roles to enhance this asymmetry (Fig. 3e).

In 1PN-ICSI-zygotes, the sp-mPN demonstrated slower histone mobility than the 1PN-ROSI, and 1PN-spt-partheno (Fig. 1f and Supplementary Fig. 3a). This result indicated that sperm actively reduced the chromatin dynamics. In Fig. 1i and Supplementary Fig. 6, although NaOH-treated inactivated sperm lost the ability to activate the oocyte, the ability to reduce histone mobility was retained, indicating that the unidentified sperm-derived chromatin dynamics reducing factors are not associated with the sperm surface. A recent broadly accepted theory states that sperm are more than mere vehicles to carry the paternal haploid genome into the oocyte. Indeed, sperm carried huge kinds of RNA into oocytes at fertilization. Several studies showed that during epididymal transit from the testis to the cauda epididymis; sperm obtained a small RNA payload[12,32]. Since tICSI-zygotes, which have no such RNA payload, exhibit parental asymmetric chromatin dynamics, the RNA payload of the mature sperm might not be involved in the decreasing histone mobility of the zygotic chromatin. However, the possibility that sperm RNA is involved in the paternal repressive effect on the chromatin structure must still be considered. Reportedly, sperm RNA is deeply embedded in the sperm head[33]. Such RNA is not easy to extract, and NaOH treatment did not completely dissolve the sperm head, indicating that the RNA was not eliminated[34]. In addition to the embedded RNA, proteins are possibly the responsible factors. Following the initiation of spermiogenesis, during which there is no transcription, specific stored RNAs were translated to proteins[19]. Since our results indicated that the ability of sperm to reduce histone mobility is acquired after the initiation of spermiogenesis, it is also possible that newly synthesized proteins at this phase are responsible for the reduction of histone mobility. To understand the importance and mechanisms of parental epigenetic competition, further studies are needed to test this hypothesis and identify the RNA and/or protein molecules responsible for zygotic chromatin dynamics.

Round spermatids do not harbor the ability to reduce histone mobility (Fig. 1f and Supplementary Fig. 3a). Moreover, when compared, in 1PN-zygotes, rs-mPNs and fPNs, which are derived from transferred meiotic spindles, exhibited the same level of histone mobility. However, there was significant asymmetric histone mobility of the parental PNs in ROSI-zygotes (Fig. 1i, Supplementary Fig. 6). Our ROSI-zygote production strategy employed a post-activation protocol[35] to improve the rate of 2PN formation[6]. In this protocol, the oocytes injected with round spermatids were activated within 30 min after ROSI. As a result, the round spermatid genome was able to avoid premature chromatin condensation followed by extrusion of the pseudo polar body. Thus, chromosome condensation and incorporation

of maternal factors did not seem to be equal between the rs-mPN and fPN, suggesting the possibility that the rs-mPN harbored more histone mobility-promoting factors than the fPN. Collectively, these findings suggest that it is probable that the differences in dynamics during PN formation contribute to the parental epigenetic competition.

The results of this study indicated that the sperm or sp-mPN exerted an ability to reduce histone mobility in both parental PNs. This finding raises the question of the biological significance of the reduction of zygotic chromatin dynamics by sperm. Bui et al.[36] reported that sperm have the ability to regulate transcriptional activity. Thus, sperm play an important role in the regulation of zygotic genome activation (ZGA). The reduction of histone mobility by sperm might be involved in the regulation of ZGA. It was thought that promiscuous transcription occurs during minor ZGA and correlates with extensive chromatin dynamics by FRAP[37]. Our results demonstrated that sperm-derived repressive factors condense the paternal chromatin structure in the sp-mPN and then the extent of histone mobility becomes comparable to that of the rs-mPN, indicating that in the absence of sperm-derived chromatin condensing factors, the paternal chromatin structure derived from the sperm should be extremely dynamic. Then, it is possible that such an extreme chromatin dynamics will cause abnormalities to the transcriptome during ZGA. To assess this possibility, the repressive factors must be identified with the use of a knockdown/knockout experimental system, which was not possible in the current study.

## Methods

**Animals**. Eight to 12-week-old female B6D2F1 (C57BL/6 × DBA2) ($n = 102$) and 10–14-week-old male ICR ($n = 42$) mice (SLC, Shizuoka, Japan) were used as oocyte and spermatozoa donors, respectively. All animal experiments were approved by the Ethics Committee of the University of Yamanashi (reference number: A29-24) and conducted in accordance with the Guide for the Care and Use of Laboratory Animals and the ARRIVE guidelines. All mice were housed under specific pathogen-free conditions at a constant temperature of 25 °C, relative humidity of 50%, and a 14/10-h light/dark period with ad libitum access to a commercial diet and distilled water. In this study, body weight was not measured because the body weight of young mice has no effect on embryo quality.

**ICSI and ROSI**. Obtained cumulus cells and oocyte complexes were treated with hyaluronidase for 10 min and the denuded oocytes were collected. For ICSI, spermatozoa were obtained from the cauda epididymis and then cultured in human tubal fluid[38] for capacitation. Prior to cytosolic injection of the denuded oocytes, the sperm tails were eliminated with a Piezo drive micromanipulator (Prime Tech Ltd., Ibaraki, Japan) in Chatot, Ziomek, Bavister (CZB)-HEPES medium supplemented with 10% PVP (10% PVP-CZB-HEPES)[39]. The zona pellucida and cytosolic membranes were also disrupted with a Piezo drive micromanipulator. For ROSI, ELSI, and testicular ICSI, the harvested testes were minced with scissors, sieved through a Mini Cell Strainer, and then re-suspended in 10% PVP-CZB-HEPES. The nucleus of each round spermatid was collected with a narrow pipette with a diameter of 7–8 μm. The zona pellucida and cytosolic membrane were disrupted in the same manner as for ICSI. The oocytes injected with round spermatid were activated by culturing in $Ca^{2+}$-free CZB medium containing 5 mM $SrCl_2$ for 1–2 h. The tails of the testicular sperm were also cut with a Piezo drive micromanipulator as with ICSI.

**Enucleation and injection of the nuclei of oocytes**. Freshly collected oocytes were transferred into 5 μg/μl of cytochalasin B (CB) containing HEPES-buffered CZB. After 10 min, the nuclei were aspirated with a glass capillary tube[40]. After enucleation, the ooplasm was washed and cultured in CZB until micro-insemination. In some experiments, the aspirated nuclei were injected into enucleated ooplasms or un-enucleated MII oocytes in CB containing HEPES-buffered CZB[41].

**Enucleation of male PN**. Before enucleation, the zygotes with two PNs at 7 hpi were cultured in $K^+$ Simplex Optimized Medium (KSOM)[42], containing CB for 20 min. The zygotes were then transferred into CB containing HEPES-buffered CZB. The larger PN and furthest away from the second polar body was deemed the mPN, which was aspirated from the zygote. The enucleated zygotes were washed and cultured in KSOM.

**Delay ICSI**. Collected oocytes were subjected to parthenogenetic activation in $Ca^{2+}$-free CZB medium containing 5 mM $SrCl_2$. After 1 h, the activated oocytes with extruding second polar bodies were collected for micro-insemination with capacitated spermatozoa. At 2 h after activation, the zygotes with an obvious extruded second polar body were used for micro-insemination.

**In vitro fertilization**. Spermatozoa were obtained from ICR mice. For capacitation, the spermatozoa were cultured for 1 h before insemination. Cumulus cells and oocyte complexes were obtained from super-ovulated BDF1 female mice by injection of 7.5 IU of equine chorionic gonadotropin (ASKA Pharmaceutical Co., Ltd., Tokyo, Japan) and human chorionic gonadotropin (ASKA Pharmaceutical) at 46–50-h intervals. Cumulus cells and oocyte complexes were inseminated with capacitated sperm in human tubal fluid medium supplemented with bovine serum albumin (BSA; Sigma-Aldrich Corporation, St. Louis, MO, USA) at 3 mg/ml. At 1–2 h postinsemination, the zygotes were washed and cultured in KSOM medium under a humidified atmosphere of 5% $CO_2$/95% air at 38 °C.

**Synthesis of mRNA**. The plasmid pTOPO eGFP-H2B (Ooga et al.)[1] encoding enhanced green fluorescent protein (eGFP)-fused histone H2B was linearized by *Not*1 overnight. Afterward, the plasmid was purified with phenol/chloroform and then precipitated with ethanol. Purified DNA was dissolved in nuclease-free water as template DNA for subsequent in vitro transcription with using mMESSAGE MACHINE sp6 kit (Themo Fisher Scientific, MA, USA). Synthesized mRNA was then processed with a poly A tailing kit (Themo Fisher Scientific). The mRNA with a poly A tail was purified and precipitated with lithium chloride precipitation solution, dissolved, and stored at 500 ng/µl and −80 °C until use.

**Zygotic fluorescence recovery after photobleaching (zFRAP) analysis**. mRNA encoding eGFP-H2B (250 ng/µl) was prepared as shown above and injected into the cytoplasm of unfertilized MII oocytes or zygotes at 1–2 h after insemination. We confirmed that this concentration did not produce excess amounts of eGFP-H2B on zFRAP analysis owing to the fact that expressed eGFP-H2B was not washed away through Triton X-100 treatment at almost all[1]. mRNA-injected MII oocytes were then micro-inseminated with a round spermatid, elongated spermatid, or spermatozoa. At 8 h postinsemination or -activation, the zygotes were collected for zFRAP analysis, performed as described previously[43,44], and observed under a confocal microscope (FV1200; Olympus Corporation, Tokyo, Japan). Briefly, the embryos were observed in the HEPES-buffered CZB medium covered with mineral oil on the glass bottom dish. During observations, the confocal microscope stage was warmed with a thermoplate (Tokai Hit, Shizuoka, Japan). With using Fluorview software, region of interest (ROI), reference region (ref), and background (back) were set at $40 \times 40$ pixels (7.6 µm$^2$). Before bleaching, three pictures were taken at 1.6-s intervals. Bleaching was conducted by 110-µW laser at 477-nm for 5 s. After bleaching, nine pictures were taken, and fluorescent intensities at ROI, ref, back were measured in each picture. For measuring intensity, the 477-nm laser was set at 15 µW. Relative intensity for each picture was determined as follows: the value of the fluorescent intensity at back was subtracted from those of ROI and ref. The obtained value of ROI was divided by that of ref. To calculate the recovery rate at each point, the relative intensity of ROI was divided by the average score of the relative intensity of three pictures taken before bleaching, and the recovery rates were plotted in the recovery curve[45]. Mobile fraction (MF) was calculated by using the equation as follows[46–48]: $MF = (F_{end} - F_{post})/(F_{pre} - F_{post})$, where $F_{end}$ is the relative intensity of fluorescence at the endpoint, $F_{post}$ is immediately after photobleaching, and $F_{pre}$ is before photobleaching.

**Immuno-staining**. After zFRAP analysis and observation, the zygotes were fixed with 4% paraformaldehyde containing 0.2% Triton X-100 for 20 min. After washing three times with PBS containing 1% BSA and 0.2% Tween 20, the zygotes were incubated with primary antibodies against H3K9me3 (ab8898; Abcam, Cambridge, MA, USA; 1:2000), Anti-H3.3 antibody (Clone 4H2D7, Cat# CE-040B, Cosmo Bio Co., LTD; 1:500), Anti-Flag M2 antibody (F1804, Merck, Darmstadt, Germany; 1:500), Anti-Hira (#39558, Active motif, Carlsbad, CA, USA; 1:500) diluted in PBS containing 1% BSA and 0.1% Triton X-100 at 4 °C overnight. After washing three times with PBS containing 1% BSA and 0.2% Tween 20, the zygotes were incubated with secondary antibodies (Alexa 568 conjugated anti-rabbit IgG, Alexa 488 conjugated mouse IgG; 1:500). The stained zygotes were mounted on PBS containing 4',6-diamidino-2-phenylindole or Vectashield mounting medium with DAPI (Vector laboratories, INC.). The images were obtained by using confocal microscopy FV1200.

**BrdU incorporation assay for DNA replication**. DNA replication was visualized with BrdU. IVF-derived zygotes were treated with dimethyl sulfoxide (DMSO) or Act. D from 4 hpi. BrdU was added to the medium containing DMSO or Act. D to a final concentration of 10 µM at 8 hpi. After 30 min, the zygotes were washed three times in 1% BSA/PBS and then were fixed by 3.7% PFA for 1 h at room temperature. After being washed in 1% BSA/PBS containing 0.05% Tween-20 at three times, the zygotes were incubated with 2-N HCl containing 0.1% at 37 ˚C for 1 h. After being washed in 1% BSA/PBS containing 0.1% Triton X-100, the samples

were transferred into 0.1-M Tris-HCl (pH8.5)/PBS containing 0.02% Triton X-100 for 15 min and then washed in 1%BSA/PBS. The zygotes were incubated with 1st antibodies against H3K9me3 and BrdU (Santa Cruz Biotechnology. Inc., SC51514, 1:250 in 1%BSA/PBS containing 0.2% Tween-20). After being washed in 1% BSA/ PBS containing 0.1% Tween 20. Second antibodies against mouse and rabbit IgG conjugated with Alexa 568 or Cy5 for 45 min. The stained zygotes were mounted on Vectashield mounting medium. The images were obtained using a confocal microscopy FV1000.

**5-EU (ethynyl uridine) incorporation assay for transcriptional activity**. Zygotic transcription was visualized with 5-EU. Briefly, ICSI-derived zygotes were transferred into KSOM containing 5-EU at 2 mM and cultured from 5 to 11 hpi. At 11 hpi, the zygotes were fixed by 4% PFA-0.2% Triton X-100 for 15 min and then washed in PBS containing 1% BSA and 0.2% Tween 20. Visualization of the incorporation of 5-EU with Alexa Fluor 594 was performed by following the manufacturer's instructions (Theremo Fisher, C10330). To distinguish the parental PN, the H3K9me3 antibody was used. After 5-EU labeling, the zygotes were immune-stained with H3K9me3 antibody as shown above. The stained zygotes were mounted on Vectashield mounting medium with DAPI. The images were obtained using a confocal microscopy FV1200.

**Statistical and reproducibility**. Data were shown as mean ± standard error of the mean (SEM). All statistical analyses were performed using Prism 9 software (ver. 9. 3.1, GraphPad Software, Inc., San Diego, CA, USA) with a one-way analysis of variance (ANOVA) followed by Tukey's multiple comparisons test or the paired *t*-test (for parental asymmetry analysis). A probability (*p*) value of <0.05 was considered statistically significant.

**Reporting summary**. Further information on research design is available in the Nature Research Reporting Summary linked to this article.

## Data availability

The data supporting our findings of this study was provided in Supplementary Data 1 and 2, or is also available from the corresponding author upon reasonable request.

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

## Acknowledgements

We thank Drs. F. Aoki, S. Namekawa, T. Ishiuchi, S. Funaya, Y. Fujimoto, and D. Ito, Y. Kikuchi, A. Namiki, S. Takeda, M. Nakamura, and Miss C. Yamaguchi for critical and useful comments on the manuscript. We also thank Drs. K. Yamagata and M. Tokoro for their critical comment and the kind gift of the construct of mCherry-MBD-NLS. We thank Dr. A. Inoue for the kind gift of the construct of Flag-H3.3. Financial support for this research was provided by a Grant-in-Aid for Young Scientists (grant no. 19K16012) and Research Grant for Young Scholars funded by Yamanashi Prefecture to M.O., the Asada Science Foundation, and the Canon Foundation (grant no. M20-0006) to T.W. This research was partially supported by a research project grant awarded by the Azabu University Research Services Division.

## Author contributions

M.O. and T.W. conceived and designed this study. M.O. performed most of the experiments and R.I., K.K., S.W., and S.K. performed some of the experiments. M.O., R.I., K.K., S.W., S.K., and T.W. analyzed all data. M.O. and T.W. wrote the manuscript. All authors read and edited the manuscript.

## Competing interests

The authors declare no competing interests.
