## [Peer Review File · Communications Biology]

Reviewers' comments:

Reviewer #1 (Remarks to the Author):

In this manuscript, Ooga et al study eGFP-H2B dynamics as a proxy for chromatin structure in mouse zygotes. The authors have previously observed that sperm male pronuclei (mPN) and female pronuclei (fPN) show distinct dynamics of eGFP-H2B recovery after photobleaching. In the present study, the authors show that sperm plays a role in the distinct FRAP dynamics between mPN and fPN. Using vigorous oocyte manipulation techniques that allow them to study effects of the number of PNs as well as their origin (mPN or fPN) on eGFP-H2B dynamics, the authors propose that PNs compete for "chromatin relaxation factors", resulting in different chromatin states between PNs. I think the idea of the competition model is interesting; however, I have a major concern about the interpretation of FRAP data, which would impact the main conclusion of the manuscript.

1. The authors use the mobile fraction of eGFP-H2B (MF; defined by $(F_{\text{end}} - F_{\text{post}}) / (F_{\text{pre}} - F_{\text{post}})$) as a proxy for "chromatin relaxation". However, the values of MF may be affected by the presence of free eGFP-H2B molecules in the nucleus. Indeed, the timescale of the recovery that the authors observe is the order of seconds, which is rather consistent with the diffusion of freely diffusing molecules than the slow diffusion of chromatin (e.g., Chalut et al., Biophysical Journal, 2012). In normal somatic cells, free histones get rapidly degraded. However, oocytes and early embryos of many species are provisioned with maternal histone pools and excess free histones likely accumulate in the nucleus on top of the chromatin-bound histones. Furthermore, the authors use eGFP-H2B mRNA injection, which is technically overexpression. My concern is that the differences in MF may be due to differences in the pool size of free eGFP-H2B and may not represent changes in the chromatin structure. For instance, in Figs 2A-B, PNs may compete for eGFP-H2B, not "chromatin relaxation factors", resulting in differences in eGFP-H2B dynamics in response to the change in the number of PNs. Similarly, in Figs F-G, larger nuclei may accumulate more free eGFP-H2B molecules (the amount of chromatin-bound histones is constant regardless of the nuclear size), resulting in an increase in MF values. I think the authors should closely check if there is no/negligible free eGFP-H2B in the nucleus.

2. Related to the above comment, to what extent do levels of histone/protamine composition differ between round spermatid and sperm? If the difference is significant, it could result in a different capacity for titrating eGFP-H2B onto male-PN chromatin between RS and sperm, thereby affecting the pool size of free eGFP-H2B and the MF in fPN.

3. line 254: I do not understand why Figs 3A-D (inhibitor experiments) suggest "more chromatin relaxation factors were utilized the in sp-mPN".

Additional comments:

- Because all experiments presented in this work are based on FRAP, the authors should provide sufficient details of their FRAP experiments and analysis in the Method. Please do not use "performed as described previously". I had a hard time finding the authors' previous paper that described the formula for the calculation of the mobile fraction.
- line 280: I don't think the data that the authors presented are sufficient to support this model. Additional experiments (e.g., CHX) are needed to make this claim. Otherwise, the authors should tone down the discussion.
- It is my understanding that the term "chromatin relaxation" refers to a response to DNA damage, which I think is not what the authors are studying in this manuscript. The authors may use a different term.
- line 218: ""Sr" indicates strontium." should be in the legend of (A), not (E)?

Reviewer #2 (Remarks to the Author):

There are several interesting phenomena described in Ooga, et al. manuscript. The major problem with the paper is the lack of mechanistic understanding of the relaxation vs compaction properties

of pronuclei and what biological significance either of those states impart into the zygote.

There is a high degree of technical expertise, the micromanipulations required in order to perform these experiments is incredibly impressive.

Since most of the experiments were done in the early to mid zygote stage before the first S-phase, which begins around 9 hours post-fertilization (PMID: 20442707) and the subsequent S phase will replace 50% of the histones that package each pronucleus, it is unclear whether the compaction status of the pronuclei, or the differentiation compaction status comparing the mat-PN vs pat-PN, has any functional significance for the embryo. For instance, does the compaction status affect the timing to the first mitosis, the minor wave of ZGA that occurs in the late zygote preferentially from the paternal pronucleus, or the major wave of ZGA that is executed in the 2-cell stage of the mouse embryo? The authors have established a system in which to perturb the compaction status (ICSI with 2 pat-PNs, parthenotes, etc) but they have not fully capitalized on this technical expertise to answer a meaningful biological function.

I think some of their claims are well-supported such as the effect seen in Fig 1h (and also described in lines 110-112), in which presumably the multiple pat-PNs decrease the FRAP recovery of the chromatin in both the pat-PN and the mat-PN, and this is a dose-dependent response--but no comment is made on the molecular mechanism. Is this just a simple titration of the H3.3/HIRA stores that presumably are responsible for FRAP recovery before the first S-phase and replicative histones are deposited?

There is also significant experimental detail lacking in the text and the figure legends as to what time HPF most of the experiments were performed. Since in Fig 1a they show a large effect on timing and in fig 3 they use a DNA replication inhibitor which presumably only affects S-phase timing (9HPA onwards) so we are unable to determine if Fig 3 is from 8HPA (as we might believe reading the methods) which is pre-S phase or not. If Fig 3a is from 8HPA, how do the authors explain the molecular effect of the Act D? A more explicit labeling of the figure panels themselves, the figure legends, or the text would help the reader understand the temporal aspects of these experiments better.

The authors use the terms "compaction" and "relaxation" but what they really are measuring is mobility, ie the deposition of non-bleached (newly translated?) histone into the chromatin. Both compaction/relaxation are related to mobility, for instance non-replicative histones are often deposited into sites of high histone turnover (PMID: 20508129), which by necessity need to be accessible to chaperones or remodelers, but the terms should not be used interchangeably. If the authors want to measure accessibility, there are several methods that non-epigenomicists use such as this imaging-based MNase assay (PMID: 28846101).

Reviewer #3 (Remarks to the Author):

Ooga et al investigate chromatin dynamics in zygotes, using the mouse model. Studying chromatin dynamics in early embryogenesis is important because chromatin states have significant impacts on embryo development through modulating the transcription activity. The authors have used their previously established method to quantify chromatin dynamics in paternal and maternal pronuclei (PN) and found that paternal PN brings the chromatin compaction activity to the zygote, while the oocyte cytoplasm harbors the opposing chromatin relaxation activity. The data presented in the manuscript is solid and support most of their main conclusions. However, the manuscript would greatly benefit from additional experiments and text editing.

It is striking to me that the parental asymmetry in chromatin dynamics was completely flipped when the timing of the sperm injection was delayed (Fig 2E-G). This point should be one of the main conclusions, as changing the fertilization timing was sufficient to reverse the parental asymmetry regardless of the paternal PN's compaction factors. Based on this result, I think the primary mechanism underlying parental asymmetry is the paternal PN having the advantage to start incorporating the relaxation factors first. And paternal PN's compaction factors have

supporting roles to enhance this asymmetry. I recommend the authors to reorganize the manuscript to highlight this point.

My another major point is how do the authors distinguish if it is the effect of the compaction factors or the competition for the relaxation factors that is compacting chromatin. This point was not clear to this reviewer in this manuscript. For example in Fig 2A and B, the chromatin of maternal PN became more compact when there are more maternal PN. The authors concluded that this is due to the competition for the relaxation factors between maternal PN. However, it is also possible that maternal PN (or maternal chromosomes they transferred) has compaction factors, and this is what making maternal PN more compact when there are more of them (like the experiments in Fig 1G and H where they increased the number of paternal PN). I strongly recommend the authors to clarify what makes the authors conclude if it is the effect of compaction factors or the competition for the relaxation factors that is making chromatin more compact in each experiment. Otherwise, this reviewer thinks that there are multiple other possible models to explain the data.

Additional points

- line 162: brief explanation of zFRAP should come up much earlier in the Results section when it is first used.
- line 164: This is not a specialized journal, and the authors should clearly explain the differences between ROSI, ELSI, tICSI, ICSI, and iICSI, so that the readers would understand why the authors need to perform these experiments to test the hypothesis.
- line 245: The subtitle "More chromatin relaxer was utilized in sp-mPN than fPN" is an overstatement, because what the authors have performed is treating zygotes with different RNA polymerase inhibitors and not directly working with the relaxation factors.
- line 257: To directly test the idea that the relaxation factors are supplied through the zygotic translation, it would be interesting to treat zygotes with Cycloheximide to block translation.

Response to reviewers' comments

**Reviewer #1 (Remarks to the Author):**

In this manuscript, Ooga et al study eGFP-H2B dynamics as a proxy for chromatin structure in
mouse zygotes. The authors have previously observed that sperm male pronuclei (mPN) and
female pronuclei (fPN) show distinct dynamics of eGFP-H2B recovery after photobleaching. In
the present study, the authors show that sperm plays a role in the distinct FRAP dynamics
between mPN and fPN. Using vigorous oocyte manipulation techniques that allow them to
study effects of the number of PNs as well as their origin (mPN or fPN) on eGFP-H2B
dynamics, the authors propose that PNs compete for “chromatin relaxation factors”, resulting in
different chromatin states between PNs. I think the idea of the competition model is interesting;
however, I have a major concern about the interpretation of FRAP data, which would impact the
main conclusion of the manuscript.

1. The authors use the mobile fraction of eGFP-H2B (MF; defined by $(F_{end}-F_{post})/(F_{pre}-$
$F_{post})$) as a proxy for “chromatin relaxation”. However, the values of MF may be affected by
the presence of free eGFP-H2B molecules in the nucleus. Indeed, the timescale of the recovery
that the authors observe is the order of seconds, which is rather consistent with the diffusion of
freely diffusing molecules than the slow diffusion of chromatin (e.g., Chalut et al., *Biophysical*
*Journal*, 2012). In normal somatic cells, free histones get rapidly degraded. However, oocytes
and early embryos of many species are provisioned with maternal histone pools and excess free
histones likely accumulate in the nucleus on top of the chromatin-bound histones. Furthermore,
the authors use eGFP-H2B mRNA injection, which is technically overexpression. My concern

is that the differences in MF may be due to differences in the pool size of free eGFP-H2B and
may not represent changes in the chromatin structure. For instance, in Figs 2A-B, PNs may
compete for eGFP-H2B, not “chromatin relaxation factors”, resulting in differences in eGFP-
H2B dynamics in response to the change in the number of PNs. Similarly, in Figs F-G, larger
nuclei may accumulate more free eGFP-H2B molecules (the amount of chromatin-bound
histones is constant regardless of the nuclear size), resulting in an increase in MF values. I think
the authors should closely check if there is no/negligible free eGFP-H2B in the nucleus.

Response: Thank you for your valuable suggestion. As per your suggestion, the amount
of free histone may affect histone mobility. We have recognized this importance. Therefore, in
our previous study, we confirmed the scarce free histone in the PN with detergent-treated zygotes
(**Confidential Figure for reviewer only 1, see below**) (Ooga *et al.*, 2016). Prior to fixation, the
zygotes were treated with Triton X-100, owing to this the unbound protein fraction to chromatin
was washed away (Hajkova *et al.* 2010). In our results, Triton X-100 treatment did not decrease
the amount of eGFP-H2B but eGFP protein used as control. Therefore, there is no/negligible free
eGFP-H2B in the pronucleus of our experimental condition. To clarify this, we have added the
sentence below in the materials and methods section.

**(P29 line 495-498)**

“We confirmed that this concentration did not produce excess amounts of eGFP-H2B
on zFRAP analysis owing to the fact that expressed eGFP-H2B was not washed away through
Triton X-100 treatment at almost all (Ooga *et al.* 2016).”

In addition, zygotes are known to have large nucleoli (NPB: nucleolar precursor body), and excess
of eGFP-H2B localized in the NPB. This structure does not contain genomic DNA; therefore,

there is no chromatin. When analyzed with FRAP, unusually high extreme values of histone
mobility were obtained (MF = 90%, **Confidential Figure for reviewers only 2, see below**)(Ooga
*et al.*, 2016). This suggested eGFP-H2B in the nucleoplasm forms a chromatin structure.
Altogether, we consider our zFRAP experiment to reflect the chromatin state.

Furthermore, we performed an additional experiment to examine the amount of
incorporated eGFP-H2B in 1PN-ICSI- and 1PN-SPT partheno-zygotes and found that 1PN-ICSI-
zygotes have a significantly higher amount of eGFP-H2B in the pronuclei than 1PN-SPT
(**Supplementary Figure 3b, c**). However, the former showed lower histone mobility than the
latter, demonstrating that the amount of incorporated eGFP-H2B was not a determinant of histone
mobility. We have added the following sentence to the manuscript:

**(P10 line 147-151)**

“Higher eGFP-H2B expression levels may result in higher histone mobility. We
compared eGFP-H2B expression levels between 1PN-ICSI and 1PN-partheno and confirmed that
the histone mobility was determined regardless of the expression level of eGFP-H2B
(**Supplementary Figure 3b, c**), suggesting that histone mobility reflects a chromatin state.”

Finally, the new suggestion below was added to the Results section.

**(P10 line 169-172.)**

“Notably, 1PN ICSI and 1PN-partheno have similar sized PN, but they showed distinct
histone mobilities (**Fig. 1e, f**). Furthermore, 1♂ (2sp) and 2♂ (2sp) zygotes whose PN sizes were
different, showed similar histone mobility levels (**Fig. 1g, h**), indicating that histone mobility was
determined regardless of PN size.”

2. Related to the above comment, to what extent do levels of histone/protamine composition

differ between round spermatid and sperm? If the difference is significant, it could result in a
different capacity for titrating eGFP-H2B onto male-PN chromatin between RS and sperm,
thereby affecting the pool size of free eGFP-H2B and the MF in fPN.

Response: Thank you for your valuable suggestion. RS chromatin was composed of
100% histones; however, sperm chromatin contains only 1 % owing to which it is possible that
there is a difference in the amount of eGFP-histone of male PNs in ICSI and ROSI-zygotes.
Following your suggestion, we examined the eGFP-H2B amounts and found that there was no
significant difference in the male PN of both types of zygotes. Therefore, we added the sentence
below in the Results section.

**(p11 line 177-182)**

“First, due to histone–protamine replacement during spermiogenesis, histone
protein is drastically reduced in spermatozoa (Wang et al. 2019; Rathke et al. 2014). Therefore,
it was possible that sp-mPN in ICSI-zygotes acquired the extremely higher amounts of eGFP-
H2B and affected histone mobility. However, eGFP-H2B expression levels were not different
between sp-mPN and rs-mPN (Supplementary Figure 6b, c).”

3. line 254: I do not understand why Figs 3A-D (inhibitor experiments) suggest “more chromatin
relaxation factors were utilized the in sp-mPN”.

Response: Thank you for your insightful suggestion.

All reviewers kindly pointed out the difficulties in understanding the purpose and strategy of these
inhibitors’ experiments.

Therefore, we have corrected the text for the interpretation of the results as shown in the Results

section.

**(p17 line 298–306)**

“We further examined whether other RNA-polymerase, RNA polymerase I, and III
(Pol I and Pol III), dependent transcriptions did not regulate the establishment of parental
asymmetric chromatin dynamics by using actinomycin D (Act D) and Pol III inhibitor (Pol IIIi).
Act D could block not only Pol I but also DNA polymerase; however, in our experiment, DNA
replication was not affected (**Supplementary Figure 13**). Both inhibitors induced the reduction
of histone mobility particularly in sp-mPN, suggesting that pol I and pol III mediated
transcription is involved in the higher histone mobility observed in sp-mPN (**Supplementary**
**Figure 14 a–d**).”

Additional comments:

- Because all experiments presented in this work are based on FRAP, the authors should provide
sufficient details of their FRAP experiments and analysis in the Method. Please do not use
“performed as described previously”. I had a hard time finding the authors’ previous paper that
described the formula for the calculation of the mobile fraction.

**Response:** Thank you for your suggestion. We added the sentence to explain the methods how
to conduct and analyze FRAP experiments in Materials and methods section.

**(p29 line 503–520)**

“Briefly, the embryos were observed in the HEPES-buffered CZB medium covered
with mineral oil on the glass bottom dish. During observation, the stage of the confocal
microscope was warmed with a thermo plate (Tokai Hit, Shizuoka, Japan). With using
Fluorview software, region of interest (ROI), reference region (ref), and background (back)
were set at 40×40 pixels ($7.6 \mu\text{m}^2$). Prior to bleaching, 3 pictures were taken at 1.6-s intervals.
Bleaching was performed with $110 \mu\text{W}$ laser at 477-nm for 5 s. After bleaching 9 pictures were

taken, and fluorescent intensities at ROI, ref, back were measured in each picture. For
measuring intensity, the 477-nm laser was set at 15 μ W. Relative intensity for each picture was
determined as follows: the value of the fluorescent intensity at back was subtracted from those
of ROI and ref. The obtained value of ROI was divided by that of ref. To calculate the recovery
rate at each point, the relative intensity of ROI was divided with the average score of the
relative intensity of three pictures taken before bleaching, and the recovery rates were plotted in
the recovery curve (Kimura et al. 2004). Mobile fraction (MF) was calculated by using the
equation as follows (Subramanian et al. 2013; Bae et al. 2012; Dieteren et al. 2011): $MF =$
$(F_{end}-F_{post})/(F_{pre}-F_{post})$, where F_{end} is the relative intensity of fluorescence at the endpoint,
F_{post} is immediately after photo-bleaching, and F_{pre} is before photo-bleaching.”

- line 280: I don't think the data that the authors presented are sufficient to support this model.
Additional experiments (e.g., CHX) are needed to make this claim. Otherwise, the authors should
tone down the discussion.

Response: Thank you for your suggestion. CHX use is ideal to prove that the
enhancing ability depends on zygotic translation. However, CHX also inhibits eGFP-H2B
translation from injected mRNA, which is required for our zFRAP experiment (**Confidential**
**Figure only for reviewers 3; see below**). Therefore, it is impossible to investigate histone
mobility in CHX-treated zygotes and we changed the results section and its interpretation.
**(p17 line 298–306)**

“We further examined whether other RNA-polymerase, RNA polymerase I, and III (Pol
I and Pol III), dependent transcriptions did not regulate the establishment of parental asymmetric
chromatin dynamics by using actinomycin D (Act D) and Pol III inhibitor (Pol IIIi). Act D could

block not only Pol I but also DNA polymerase, but in our experiment DNA replication was not
affected (**Supplementary Figure 13**). Both inhibitors induced the reduction of histone mobility
particularly in sp-mPN, suggesting that pol I and pol III mediated transcription is involved in the
higher histone mobility in sp-mPN (**Supplementary Figure 14 a-d**).”

- It is my understanding that the term “chromatin relaxation” refers to a response to DNA damage,
which I think is not what the authors are studying in this manuscript. The authors may use a
different term.

Response: Deeply considering and following the reviewers’ comment, we changed
“chromatin relaxation” into “chromatin dynamics” or “histone mobility” throughout the paper
depending on the context, which were used in previous studies using FRAP technique performed
by other research groups (Bošković *et al.* 2014).

- line 218: “Sr” indicates strontium.” should be in the legend of (A), not (E)?

Response: Following the reviewer’s comment, we have added the explanation of “Sr” in the
legend for Fig. 2A.

**(p15 line 269)**

“(a)Illustration of the preparation of 1, 2, and 4PN partheno-zygotes and fluorescence images of
the fPNs. “Sr” indicates strontium, which induces oocyte activation.”

**Reviewer #2 (Remarks to the Author):**

There are several interesting phenomena described in Ooga, et al. manuscript. The major problem
with the paper is the lack of mechanistic understanding of the relaxation vs compaction properties
of pronuclei and what biological significance either of those states impart into the zygote.

There is a high degree of technical expertise, the micromanipulations required in order to
perform these experiments is incredibly impressive.

Since most of the experiments were done in the early to mid zygote stage before the first S-phase,
which begins around 9 hours post-fertilization (PMID: 20442707) and the subsequent S phase
will replace 50% of the histones that package each pronucleus, it is unclear whether the
compaction status of the pronuclei, or the differentiation compaction status comparing the mat-
PN vs pat-PN, has any functional significance for the embryo. For instance, does the compaction
status affect the timing to the first mitosis, the minor wave of ZGA that occurs in the late zygote
preferentially from the paternal pronucleus, or the major wave of ZGA that is executed in the 2-
cell stage of the mouse embryo? The authors have established a system in which to perturb the
compaction status (ICSI with 2 pat-PNs, parthenotes, etc) but they have not fully capitalized on
this technical expertise to answer a meaningful biological function.

Response: Thank you for your important suggestion. We performed 5-EU experiments to reveal
the significance and effect on transcription by altered chromatin dynamics in delay ICSI-
zygotes. The results demonstrated that the chromatin dynamics are involved in the regulation of
parental asymmetric transcriptional activity. We added the data (Fig. 3), interpretation, and

modifications to materials and methods (p31 line 554–564) and results section (p17 line 282–
298), respectively.

(p31 line 554–564)

“5-EU (Ethynyl Uridine) incorporation assay for transcriptional activity

Zygotic transcription was visualized with 5-EU. Briefly, ICSI derived zygotes were
transferred into KSOM containing 5-EU at 2 mM and cultured from 5 hpi to 11 hpi. At 11 hpi,
the zygotes were fixed by 4% PFA-0.2% Triton X-100 for 15 min and then washed in PBS
containing 1% BSA and 0.2% Tween 20. Visualization of the incorporation of 5-EU with Alexa
Fluor 594 was performed by following the manufacturer’s instructions (Thermo Fisher,
C10330). To distinguish the parental PN, the H3K9me3 antibody was used. After 5-EU
labeling, the zygotes were immune-stained with H3K9me3 antibody as shown above. The
stained zygotes were mounted on Vectashield mounting medium with DAPI. The images were
obtained by using confocal microscopy FV1200.”

(p17 line 282–298)

“*Chromatin dynamics are involved in the regulation of the parental asymmetric*
*transcriptional activity.*”

High histone mobility is anticipated to be involved in the zygotic transcriptional
permissive state (Du et al. 2021; Xia and Xie 2020). sp-mPN has higher transcriptional activity
possibly due to a more transcriptional permissive chromatin state than fPN (Schultz 2002).
Therefore, we hypothesized that the higher histone mobility in sp-mPN is important for
regulating parental asymmetric transcriptional activities. To examine this, we took an advantage
of our discovery that delay-ICSI induced the reversed parental asymmetric chromatin dynamics
(Fig. 2e, f, and g). As expected, the longer delay reduced transcriptional activity in sp-mPN,
and eventually reversed parental asymmetric transcription was observed (Fig. 3a, b).
Considering that eviction and deposition of histone H2B could be coupled with transcription, it

is possible that lowered transcriptional activity caused slower histone mobility. Nevertheless,
the treatment with alpha-amanitin (ama), which is a RNA-polymerase II (Pol II) inhibitor, led to
no reduction of histone mobility (Fig. 3c, d). These results suggest that chromatin dynamics are
involved in the regulation of the pol II-mediated parental asymmetric transcriptional activity.”

I think some of their claims are well-supported such as the effect seen in Fig 1h (and also
described in lines 110-112), in which presumably the multiple pat-PNs decrease the FRAP
recovery of the chromatin in both the pat-PN and the mat-PN, and this is a dose-dependent
response--but no comment is made on the molecular mechanism. Is this just a simple titration of
the H3.3/HIRA stores that presumably are responsible for FRAP recovery before the first S-
phase and replicative histones are deposited?

Response: Thank you for this important suggestion. We examined endogenous and exogenous
“H3.3” levels and “Hira” in the parental PNs of ROSI- and ICSI-zygotes as you suggested. The
results were added to Supplementary Figure 7, and we added the sentence shown below.

(p12 line 200–207)

“A variant of histone H3, H3.3, is considered important to establish open chromatin
structure in parental PN after fertilization (Ishiuchi et al. 2021; Akiyama et al. 2011; Koh et al.
2010). Therefore, we examined whether the differences of titration capacity of H3.3/Hira (H3.3
specific chaperon) were correlated with the chromatin dynamics in parental PNs of ROSI-, and
ICSI-zygotes. However, there was no correlation between histone mobility and each level of
endogenous or exogeneous H3.3 and Hira, thereby indicating that it could not explain the levels
of histone mobility in parental PNs in ROSI- and ICSI-zygotes by H3.3 level (Supplementary
Figure 7).”

There is also significant experimental detail lacking in the text and the figure legends as to what
time HPF most of the experiments were performed. Since in Fig 1a they show a large effect on
timing and in fig 3 they use a DNA replication inhibitor which presumably only affects S-phase
timing (9HPA onwards) so we are unable to determine if Fig 3 is from 8HPA (as we might
believe reading the methods) which is pre-S phase or not. If Fig 3a is from 8HPA, how do the
authors explain the molecular effect of the Act D? A more explicit labeling of the figure panels
themselves, the figure legends, or the text would help the reader understand the temporal
aspects of these experiments better.

Response: Thank you for your suggestion.

Reviewers kindly pointed out the difficulties to understand the purpose and strategy of these
inhibitor experiments. Owing to the lower specificity of actinomycin D (actD), the interpretation
of this experiment with act D became very difficult. We entirely modified the corresponding result
section and legend for Supplementary Figure 14c

**(p17 line 298–306)**

“We further examined whether other RNA-polymerase, RNA polymerase I, and III (Pol
I and Pol III), dependent transcriptions did not regulate the establishment of parental asymmetric
chromatin dynamics by using actinomycin D (Act D) and Pol III inhibitor (Pol IIIi). Act D could
block not only Pol I but also DNA polymerase; however, in our experiment, DNA replication was
not affected (**Supplementary Figure 13**). Both inhibitors induced the reduction of histone
mobility particularly in sp-mPN, suggesting that pol I and pol III mediated transcription is
specifically involved in the higher histone mobility in sp-mPN (**Supplementary Figure 14 a–d**).”

**(p62 line 879–882)**

“(c) Recovery curve of IVF-zygotes treated with 20 μ M Pol IIIi. Control zygotes were
treated with 0.1% DMSO. Treatment with these inhibitors started soon after mRNA
microinjection (around 2 hpi). zFRAP analysis was started at 8 hpi.

The authors use the terms "compaction" and "relaxation" but what they really are measuring is
mobility, i.e. the deposition of non-bleached (newly translated?) histone into the chromatin. Both
compaction/relaxation are related to mobility, for instance non-replicative histones are often
deposited into sites of high histone turnover (PMID: 20508129), which by necessity need to be
accessible to chaperones or remodelers, but the terms should not be used interchangeably. If the
authors want to measure accessibility, there are several methods that non-epigenomicists use such
as this imaging-based MNase assay (PMID: 28846101).

Response: Thank you for your important suggestion. Deeply considering and following the
reviewers' comment, we changed “chromatin relaxation” into “chromatin dynamics” or “histone
mobility” throughout the paper depending on the context. Both “chromatin dynamics” and “histone
mobility” were used in the previous studies with FRAP that were performed by other research
groups (Bošković *et al.* 2014).

**Reviewer #3 (Remarks to the Author):**

Ooga et al investigate chromatin dynamics in zygotes, using the mouse model. Studying
chromatin dynamics in early embryogenesis is important because chromatin states have
significant impacts on embryo development through modulating the transcription activity. The
authors have used their previously established method to quantify chromatin dynamics in paternal

and maternal pronuclei (PN) and found that paternal PN brings the chromatin compaction activity
to the zygote, while the oocyte cytoplasm harbors the opposing chromatin relaxation activity. The
data presented in the manuscript is solid and support most of their main conclusions. However,
the manuscript would greatly benefit from additional experiments and text editing.

It is striking to me that the parental asymmetry in chromatin dynamics was completely flipped
when the timing of the sperm injection was delayed (Fig 2E-G). This point should be one of the
main conclusions, as changing the fertilization timing was sufficient to reverse the parental
asymmetry regardless of the paternal PN's compaction factors. Based on this result, I think the
primary mechanism underlying parental asymmetry is the paternal PN having the advantage to
start incorporating the relaxation factors first. And paternal PN's compaction factors have
supporting roles to enhance this asymmetry. I recommend the authors to reorganize the
manuscript to highlight this point.

Response: Thank you for your very kind and helpful suggestion.

Considering the reviewer's comment, we changed the title into “**Parental competition for the**
regulators of chromatin dynamics in mouse zygotes,” and modified the abstract section
entirely.

(p2 line 20–34)

**Abstract**

The underlying mechanism for parental asymmetric chromatin dynamics is still
unclear. To reveal this, we investigated chromatin dynamics in parthenogenetic, androgenic, and
several types of male germ cells-fertilized zygotes with single to quad pronuclei. Here we
illustrated that parental conflicting role mediates the regulation of chromatin dynamics. Sperm
reduces chromatin dynamics in both parental PNs. During spermiogenesis, male germ cells
acquire this reducing ability and its resistance. On the other hand, oocytes can increase
chromatin dynamics. Notably, the oocytes-derived chromatin dynamics enhancing ability is

dominant for the sperm-derived opposing one. This maternal enhancing ability was competed
between parental pronuclei. Delayed fertilization timing was critical for this competition and
compromised parental asymmetric chromatin dynamics and zygotic transcription. Together,
parental competition for the maternal factor enhancing chromatin dynamics is a determinant to
establish parental asymmetry, and paternal repressive effects have supporting roles to enhance
asymmetry.

We also modified the introduction section and added the sentence to impress the meanings of
parental competition.

**(p4 line 71–75)**

“In addition to the factors carried by sperm, differences in the pronuclear formation
process are critical for establishing chromatin structures in the zygotes. Soon after fertilization,
only the sperm genome undergoes protamine-histone replacement. This sperm-derived genome-
specific phenomenon possibly confers the chance to male genomes for contact by the maternally
supplied factors (Schultz 2002).”

We added the sentence below to the Results section to respond to your suggestion.

**(p14 line 264–280)**

“Notably, delaying fertilization timing was sufficient to reverse the parental
asymmetry regardless of sperm deriving ability to decrease histone mobility, suggesting that the
histone mobility promoting factors were predominant in the regulation of parental asymmetric
chromatin dynamics.”

Furthermore, we modified the Discussion section.

**(p21 line 327–329)**

“Notably, this competition is critical to the establishment of parental asymmetric
chromatin dynamics (Fig. 2e-g) and is involved in the regulation of parental asymmetric
transcriptional activity (Fig. 3a-d).”

**(p22 line 354-359)**

“Additionally, changing the fertilization timing was sufficient to reverse the parental
asymmetry regardless of the paternal repressive effects on chromatin dynamics suggest that
primary mechanism underlying parental asymmetry is the paternal PN having the advantage to
start incorporating the chromatin dynamics promoting factors first. And paternal repressive
effects have supporting roles to enhance this asymmetry (Fig. 3e).”

My another major point is how do the authors distinguish if it is the effect of the compaction
factors or the competition for the relaxation factors that is compacting chromatin. This point
was not clear to this reviewer in this manuscript. For example in Fig 2A and B, the chromatin of
maternal PN became more compact when there are more maternal PN. The authors concluded
that this is due to the competition for the relaxation factors between maternal PN. However, it is
also possible that maternal PN (or maternal chromosomes they transferred) has compaction
factors, and this is what making maternal PN more compact when there are more of them (like
the experiments in Fig 1G and H where they increased the number of paternal PN). I strongly
recommend the authors to clarify what makes the authors conclude if it is the effect of
compaction factors or the competition for the relaxation factors that is making chromatin more
compact in each experiment. Otherwise, this reviewer thinks that there are multiple other
possible models to explain the data.

Response: Thank you for your important suggestion. We reconsidered the possibility
of the involvement of chromatin repressive effect derived from fPN. In figure 1h, increasing sp-

mPN disrupted parental asymmetry. On the contrary increasing fPN did not (**Supplementary**
**Figure 5**). Furthermore, MF of 2 ♂+1 ♀ (2sp) was around 10 (**Fig. 1h**) but, that of 1 ♂+2 ♀
(2sp) was around 15-20 (**Supplementary Figure 5**), indicating that the lower ability to decrease
histone mobility of fPN.

Furthermore, in figure 1i, we compared ROSI, ELSI, tICSI and ICSI-zygotes. The
results showed that histone mobility in female PN was drastically changed depending on their
partner male germ cells. In contrast, rs-mPN and sp-mPN showed no significant changes in
histone mobility despite their differences in resistance to the ability for decreasing histone
mobility (**Supplementary Figure 9, 10**).

Collectively these results indicated that if there are chromatin compaction factors
derived from fPN, it would have little impact on parental asymmetry. Therefore, we modified
and added the sentence below to the result section.

**(p10 line 161–166)**

Thus, the sp-mPN exhibited innate resistance to the **ability for decreasing histone mobility**. The
zygotes harboring two fPNs with a single sp-mPN (“1♂ + 2♀”) still exhibited the parental
asymmetric pattern (**Supplementary Figure 5**). Thus, the additional chromatin from the fPN
failed to disrupt the parental asymmetric pattern, **indicating the scarce or little repressive effect**
**on histone mobility by fPN**.

**(p11 line 187–189)**

Furthermore, fPN in ROSI-zygotes showed higher histone mobility than ICSI-zygotes,
**indicating the scarce or little repressive effect on histone mobility by fPN**.

Additional points

- line 162: brief explanation of zFRAP should come up much earlier in the Results section
when it is first used.

Response: Thank you for your suggestion.

We corrected it as shown in p6 line 98, which is a very earlier place in the Results section.

**(p6 line 92–100)**

**“Results**

***Sperm represses chromatin dynamics in both parental PNs.***

We previously reported that asymmetric **dynamics** of the parental chromatin were
established in the late zygotic stage of the embryo at 10–12 h post insemination (hpi) (Ooga et
al., 2016). **Firstly**, we confirmed the reproducibility of **asymmetric chromatin dynamics** in the
parental chromatin ($\text{♂} > \text{♀}$) in zygotes obtained by in vitro fertilization (IVF) (**Supplementary**
**Figure 1**) and ICSI **by zygotic fluorescence recovery after photobleaching (zFRAP), which**
**revealed histone mobility (an indicator of chromatin dynamics) (Meshorer et al. 2006; Ooga et**
**al. 2016; Bošković et al. 2014).”**

- line 164: This is not a specialized journal, and the authors should clearly explain the
differences between ROSI, ELSI, tICSI, ICSI, and iICSI, so that the readers would understand
why the authors need to perform these experiments to test the hypothesis.

Response: We added a graphical visualization to show spermiogenesis in testis and cauda
epididymis (Supplementary Figure 6a) and its detailed explanation in the legend for this (p50
line 788–795). Furthermore, some modifications were also added to Fig. 1i.

**(p50 line 788–795)**

“(a) During spermiogenesis, which is the post-meiotic stage, male germ cells change the
morphology dynamically and differentiate from round spermatid (RS) to elongating spermatid
(ES) and testicular sperm (tSP) in the testis. After differentiation to tSP, they move to cauda
epididymis from the testis and then mature to sperm (SP). Micro-insemination with this SP is
called ICSI, and with round spermatid, elongating spermatid in the testis is as ROSI and ELSI
(elongating spermatid injection), respectively. We referred to microinjection with immature tSP
as “testicular ICSI” (tICSI).”

To explain the purpose of the experiment with inactivated sperm, we added the explanation
sentence below to the result section and a simple illustration (**Supplementary Figure 6f**).

**(p11 line 190–195)**

“During spermiogenesis, sperm acquires the ability to induce oocytes activation, which leads
to the initiation of the zygotic cell cycle from meiotically arrested MII oocytes after fertilization
and this ability can be easily inactivated by alkali treatment (**Supplementary Figure 6f**)(Li et
al. 2009). To examine whether this ability was involved in the parental asymmetric chromatin
dynamics,”

- line 245: The subtitle “More chromatin relaxer was utilized in sp-mPN than fPN” is an
overstatement, because what the authors have performed is treating zygotes with different RNA
polymerase inhibitors and not directly working with the relaxation factors.

Response: We modified the subtitle into “*Chromatin dynamics are involved in the*
*regulation of the parental asymmetric transcriptional activity.*” (p17 line 282-283)

- line 257: To directly test the idea that the relaxation factors are supplied through the zygotic
translation, it would be interesting to treat zygotes with Cycloheximide to block translation.

Response: Thank you for your important suggestion. We tried to conduct the experiment
using CHX to examine whether the zygotic translation is involved in chromatin loosening in
zygotes. However, our FRAP experimental system required zygotic translation to express eGFP-
H2B. Indeed, CHX treatment completely inhibits the expression of eGFP-H2B (**Confidential**
**Figure for reviewers only 3; see below**). Therefore, we could not examine the hypothesis and
removed the discussion for the involvement of the zygotic translation.

**Confidential Figures for reviewer only**

The 2 figures below are cited from our previous study (Ooga *et al.* 2016).

**Confidential figure for reviewers only 1 (from Ooga et al 2016)**

Supplemental Figure. S2

Supplemental Fig. S2. Assessment of the unbound fraction of eGFP-H2B in 1- and 2-cell stage embryos.

cRNA encoding eGFP-H2B or eGFP was injected into the cytoplasm of embryos 2 h post-insemination (hpi). One-cell (**A**) and 2-cell (**B**) stage embryos were collected at 10 and 30 hpi, respectively. After the unbound fraction of eGFP or eGFP-H2B was washed away using 0.2% Triton X-100 (see Materials and Methods), the embryos were fixed and observed for fluorescence. A single experiment, in which at least 5 embryos were examined in each group, was conducted. Representative images are shown. Since there was no embryo in which male and female pronuclei were located at a single focal plane, two images were taken for each embryo at the plane at which the male or female pronuclei was clearly observed. Bar = 20 μ m.

Supplemental Figure. S3

A

B

Supplemental Fig. S3. Mobility of eGFP-H2B in the nucleolar precursor body (NPB)

cRNA encoding eGFP-H2B was injected into the cytoplasm of 1-cell stage embryos 2 h post-insemination (hpi). Mobility of eGFP-H2B at the NPB and euchromatin region in the nucleus of 1-cell stage embryos was examined using FRAP between 10-13 hpi. The recovery curve and mobile fraction are shown on the left and right, respectively. Three independent experiments were performed and the data were accumulated. In total, more than 17 pronuclei were examined. Bar = 10 μ m

**Confidential figure for reviewers only 3** (Additional experiment conducted in this study)

In vitro fertilized-zygotes were microinjected with mRNA encoding eGFP-H2B were prepared.
After microinjection, the zygotes were transferred to cycloheximide (CHX) containing
medium. Nine hours after fertilization, fluorescent signals of eGFP-H2B were not observed in
CHX-treated zygotes. The upper panel indicates the merge of bright field and fluorescence.
The lower panel indicates bright fields.

**References**

Akiyama T, Suzuki O, Matsuda J, Aoki F. 2011. Dynamic replacement of histone H3 variants
reprograms epigenetic marks in early mouse embryos. *PLoS Genet* **7**.

Bae J, Sung BH, Cho IH, Song WK. 2012. F-actin-dependent regulation of NESH dynamics in
rat hippocampal neurons. *PLoS One* **7**: 4–15.

Bošković A, Eid A, Pontabry J, Ishiuchi T, Spiegelhalter C, Raghu Ram EVS, Meshorer E,
Torres-Padilla ME. 2014. Higher chromatin mobility supports totipotency and precedes
pluripotency in vivo. *Genes Dev* **28**: 1042–1047.

Dieteren CEJ, Willems PHGM, Swarts HG, Fransen J, Smeitink JAM, Koopman WJH,
Nijtmans LGJ. 2011. Defective mitochondrial translation differently affects the live cell
dynamics of complex i subunits. *Biochim Biophys Acta - Bioenerg* **1807**: 1624–1633.
<http://dx.doi.org/10.1016/j.bbabi.2011.09.013>.

Du Z, Zhang K, Xie W. 2021. Epigenetic Reprogramming in Early Animal Development. *Cold*
*Spring Harb Perspect Biol* a039677.

Hajkova P, Jeffries SJ, Lee C, Miller N, Jackson SP, Surani MA. 2010. Genome-wide
reprogramming in the mouse germ line entails the base excision repair pathway. *Science*
*(80-)* **329**: 78–82.

Ishiuchi T, Abe S, Inoue K, Yeung WKA, Miki Y, Ogura A, Sasaki H. 2021. Reprogramming
of the histone H3.3 landscape in the early mouse embryo. *Nat Struct Mol Biol* **28**: 38–49.
<http://dx.doi.org/10.1038/s41594-020-00521-1>.

Kimura H, Hieda M, Cook PR. 2004. Measuring Histone and Polymerase Dynamics in Living
Cells. *Methods Enzymol* **375**: 381–393.

Koh FM, Sachs M, Guzman-Ayala M, Ramalho-Santos M. 2010. Parallel gateways to
pluripotency: Open chromatin in stem cells and development. *Curr Opin Genet Dev* **20**:
492–499.

Li C, Mizutani E, Ono T, Wakayama T. 2009. Production of normal mice from spermatozoa
denatured with high alkali treatment before ICSI. *Reproduction* **137**: 779–792.

Meshorer E, Yellajoshula D, George E, Scambler PJ, Brown DT, Misteli T. 2006.
Hyperdynamic plasticity of chromatin proteins in pluripotent embryonic stem cells. *Dev*
*Cell* **10**: 105–116.

Ooga M, Fulka H, Hashimoto S, Suzuki MG, Aoki F. 2016. Analysis of chromatin structure in
mouse preimplantation embryos by fluorescent recovery after photobleaching. *Epigenetics*
**11**.

Rathke C, Baarends WM, Awe S, Renkawitz-Pohl R. 2014. Chromatin dynamics during
spermiogenesis. *Biochim Biophys Acta - Gene Regul Mech* **1839**: 155–168.
<http://dx.doi.org/10.1016/j.bbagr.2013.08.004>.

Schultz RM. 2002. The molecular foundations of the maternal to zygotic transition in the
preimplantation embryo. *Hum Reprod Update* **8**: 323–331.

Subramanian V, Mazumder A, Surface LE, Butty VL, Fields PA, Alwan A, Torrey L, Thai KK,
Levine SS, Bathe M, et al. 2013. H2A.Z Acidic Patch Couples Chromatin Dynamics to
Regulation of Gene Expression Programs during ESC Differentiation. *PLoS Genet* **9**.

Wang T, Gao H, Li W, Liu C. 2019. Essential Role of Histone Replacement and Modifications
in Male Fertility. *Front Genet* **10**: 1–15.

Xia W, Xie W. 2020. Rebooting the Epigenomes during Mammalian Early Embryogenesis.
*Stem Cell Reports* **15**: 1158–1175. <https://doi.org/10.1016/j.stemcr.2020.09.005>.

REVIEWERS' COMMENTS:

Reviewer #1 (Remarks to the Author):

The authors have satisfactorily addressed my concerns.

Reviewer #2 (Remarks to the Author):

The authors have improved some of the description of timing, techniques, etc that made the original manuscript difficult to understand.

The authors have made an effort to address my initial comment regarding the function of increased histone mobility in the zygote by including some experiments with RNA pol-II inhibitors (new figure 3c/d) and have included quantification of RNA transcription levels in the pronuclei in their system (Fig 3a/b).

The authors then conclude "Chromatin dynamics are involved in the regulation of the parental asymmetric transcriptional activity" in the figure legend for Fig 3. It is unclear what the word "involved" means. Since the authors only included the RNA pol1/III inhibitor experiments in Sup Fig 14, this main figure #3 contains mostly negative data. I think the paper would be more impactful if the authors could modify their current Fig3 to include data from Sup Fig 14.

Reviewer #3 (Remarks to the Author):

The authors have addressed to all my concerns, and therefore, I think the manuscript is now ready for publication.

Response to Reviewer's comment

The authors then conclude "Chromatin dynamics are involved in the regulation of the parental asymmetric transcriptional activity" in the figure legend for Fig 3. It is unclear what the word "involved" means. Since the authors only included the RNA polII/III inhibitor experiments in Sup Fig 14, this main figure #3 contains mostly negative data. I think the paper would be more impactful if the authors could modify their current Fig3 to include data from Sup Fig 14.

Thank you for your valuable suggestion.

As you have pointed out, "involved" is ambiguous. Therefore, we have changed the word into "**Play a regulatory role for the establishment**" (line 244), "**positively correlated**" (line 256) and "**positively regulates**" (line 270).

Furthermore, we rearranged figure 3 to include the data from Supplementary figure 14 where Act D and pol III inhibitor were used in the experiment.

We believe that our data with amanitin was probably regarded as negative since amanitin treatment has no effect but Act D and pol III inhibitor affected chromatin dynamics. However, we suggest that chromatin dynamics play an upstream role in the transcriptional activity regulation. We interpreted the result of the revised fig. 3a/b that loss of chromatin dynamics caused the decrease of transcriptional activity in male PN (or reversed parental asymmetric chromatin dynamics caused reversed parental asymmetric transcriptional activity.). Importantly, in fig. 2g, the chromatin dynamics were reversed by only a 1h delay but in fig. 3b it took 2h to completely reverse the parental asymmetry. Therefore, we could not prove that chromatin dynamics were a governing regulatory upstream factor for transcriptional activity. Taken together, we concluded that chromatin dynamics positively regulate zygotic transcriptional activity, though not as a sole factor.